# Anti-Platelet Properties of Phenolic and Nonpolar Fractions Isolated from Various Organs of *Elaeagnus rhamnoides* (L.) A. Nelson in Whole Blood

**DOI:** 10.3390/ijms22063282

**Published:** 2021-03-23

**Authors:** Bartosz Skalski, Joanna Rywaniak, Aleksandra Szustka, Jerzy Żuchowski, Anna Stochmal, Beata Olas

**Affiliations:** 1Department of General Biochemistry, Faculty of Biology and Environmental Protection, University of Lodz, Pomorska 141/143, 90-236 Łódź, Poland; bartosz.skalski@biol.uni.lodz.pl; 2Department of Immunology and Infectious Biology, Institute of Microbiology, Biotechnology and Immunology, Faculty of Biology and Environmental Protection, University of Lodz, Banacha 12/16, 90-237 Łódź, Poland; joanna.rywaniak@biol.uni.lodz.pl; 3Department of Cytobiochemistry, Faculty of Biology and Environmental Protection, University of Lodz, Pomorska 141/143, 90-236 Łódź, Poland; aleksandra.szustka@biol.uni.lodz.pl; 4Department of Biochemistry and Crop Quality, Institute of Soil Science and Plant Cultivation—State Research Institute, Krańcowa 8, 24-100 Puławy, Poland; jzuchowski@iung.pulawy.pl (J.Ż.); asf@iung.pulawy.pl (A.S.)

**Keywords:** sea buckthorn, flow cytometry, T-TAS, haemostasis, electrophoresis

## Abstract

Sea buckthorn (*Elaeagnus rhamnoides* (L.) A. Nelson) is a shrub growing in coastal areas. Its organs contain a range of bioactive substances including vitamins, fatty acids, various micro and macro elements, as well as phenolic compounds. Numerous studies of sea buckthorn have found it to have anticancer, anti-ulcer, hepatoprotective, antibacterial, and antiviral properties. Some studies suggest that it also affects the hemostasis system. The aim of the study was to determine the effect of six polyphenols rich and triterpenic acids rich fractions (A–F), taken from various organs of sea buckthorn, on the activation of blood platelets using whole blood, and to assess the effect of the tested fractions on platelet proteins: fraction A (polyphenols rich fraction from fruits), fraction B (triterpenic acids rich fraction from fruits), fraction C (polyphenols rich fraction from leaves), fraction D (triterpenic acids rich fraction from leaves), fraction E (polyphenols rich fraction from twigs), and fraction F (triterpenic acids rich fraction from twigs). Hemostasis parameters were determined using flow cytometry and T-TAS (Total Thrombus-formation Analysis System). Additionally, electrophoresis was performed under reducing and non-reducing conditions. Although all tested fractions inhibit platelet activation, the greatest anti-platelet activity was demonstrated by fraction A, which was rich in flavonol glycosides. In addition, none of the tested fractions (A–F) caused any changes in the platelet proteome, and their anti-platelet potential is not dependent on the P2Y12 receptor.

## 1. Introduction

Anticoagulant and antiplatelet drugs play an important role in the prevention and treatment of cardiovascular thrombotic events caused by various mechanisms. However, these drugs may also induce various side effects, which are well described [1]. In addition, certain dietary components, including phenolic compounds and supplements with antiplatelet properties may also reduce platelet activation, and may have a significant influence on the prophylaxis and treatment of cardiovascular diseases (CVDs) [2,3,4]. Some of the most promising sources of active compounds for the prevention and treatment of CVDs may be the organs and fruits of sea buckthorn (*Elaeagnus rhamnoides* (L.) A. Nelson) [5,6,7].

The medicinal values of sea buckthorn organs, especially the fruits, have been confirmed both by traditional medicine and by scientific reports [4,8]. Our previous in vitro findings, based on the analysis of washed blood platelets and platelet-rich plasma, demonstrate that both polyphenols rich and triterpenic acids rich fractions from sea buckthorn leaves and twigs modulate the coagulation process in human plasma and modulate human platelet function [9,10]. In addition, our earlier results indicate that both types of fraction from sea buckthorn fruits possess anticoagulant and antioxidant properties [4]. However, their mechanisms of action in whole blood are not known. Therefore, the aim of the present in vitro study was to determine the anticoagulant and anti-platelet properties of selected polyphenols rich and triterpenic acids rich fractions isolated from different parts of *Elaeagnus rhamnoides* (L.) A. Nelson.

Six fractions were studied: fraction A—polyphenols rich fraction from fruits, rich in non-acylated and acylated flavonoids and nonpolar compounds; fraction B—triterpenic acids rich fraction from fruits; fraction C—polyphenols rich fraction from leaves; fraction D—triterpenic acids rich fraction from leaves; fraction E—polyphenols rich fraction from twigs; fraction F—triterpenic acids rich fraction from twigs. The analysis was performed in whole blood using flow cytometry and total thrombus-formation analysis system (T-TAS). The effect of the plant fractions on blood platelet function was determined by flow cytometry, i.e., by quantifying the cell-surface expression of the platelet activation markers P-selectin and GPIIb/IIIa, both in unstimulated platelets and in those treated with the agonists ADP (adenosine diphosphate) and collagen. Blood platelet function was also assessed based on vasodilator-stimulated phosphoprotein (VASP) phosphorylation in blood platelets. The aim of our study was also to identify changes in blood platelet proteomes following treatment with sea buckthorn fractions A–F.

## 2. Results

The results indicated altered blood platelet activation states in all samples treated with tested plant fractions (A–F) compared with platelets (without plant fractions); this was true for both samples treated with agonist (ADP or collagen) and resting platelets (Figure 1, Figure 2 and Figure 3). However, these changes were not always statistically significant. Treatment with fraction C significantly reduced the expression of CD62P by about 30% for 5 µg/mL, and about 20% for 50 µg/mL in resting blood platelets (Figure 1A).

PAC-1 binding was found to be reduced by fractions A, B, and C, but increased by fraction F (Figure 2A). Moreover, fraction A was found to reduce PAC-1 binding in collagen-activated blood platelets when administered at both 5 µg/mL and 50 µg/mL, with a 30% reduction observed at 50 µg/mL (Figure 2D).

No differences were observed between the electropherograms of blood platelets treated with plant fractions (A–F) and the control samples (Figure 4). Nor were any differences observed between the PRI values of the samples treated with 50 µg/mL of the plant fractions (A–F) and the control samples (Figure 5).

Changes were also observed in the AUC_10_ measured by T-TAS (Figure 6). Four of the tested plant fractions (A, C, D, and E) markedly decreased AUC_10_ relative to control when administered at 50 µm/mL. However, no change was observed for the other two fractions (B and F) administered at the same concentration (Figure 6).

A comparison of the effects of all six tested fractions (A–F) at the highest used concentration (50 µg/mL) on selected biomarkers of platelet activation, as measured by cytometric analysis and T-TAS, is given in Table 1. Of the six extracts, fraction A, i.e., the phenolic fraction from fruits, had the strongest anti-platelet potential. Fraction A inhibited PAC-1 expression in three used models: (1) non-activated platelets, (2) platelets activated by 10 µM ADP, (3) platelets activated by 10 µg/mL collagen. This fraction also demonstrated anti-coagulant potential, measured by T-TAS (Table 1).

Table 1 also shows comparative effect of six tested fractions (A–F; 50 µg/mL) and aronia berry extract (50 µg/mL) on the selected parameters of platelet activation. 

## 3. Discussion

Shifts in hemostasis, including changes in blood platelet function, are well documented in cardiovascular diseases. A number of currently-used clinical antithrombotic agents, such as aspirin, are known for having side effects that may cause serious hematological risk, abnormality or gastrointestinal damage [11]. Therefore, plant preparations may represent an attractive alternative for antithrombotic agents as they demonstrate anti-platelet activity, and sometimes antioxidant potential, but without the side effects associated with artificial preparations.

To determine the influence of such natural preparations on selected aspects of hemostasis, such as platelet activation, the present study used a combination of flow cytometry and T-TAS to examine the effect of treating whole blood samples with six plant fractions isolated from different organs of *Elaeagnus rhamnoides* (L.) A. Nelson. T-TAS, a microchip-based flow chamber system that evaluates thrombogenicity in whole blood, may also be used to evaluate the influence of anti-thrombotic preparations on blood platelet activation and coagulation reactions over a collagen or collagen/tissue thromboplastin-coated surface [12]. The present experiment used surfaces coated with collagen for visualizing platelet thrombus formation in the presence of six tested plant fractions (A–F), four of which (A, C, D, and E) were found to demonstrate anti-coagulant potential.

Platelet activation was also assessed by flow cytometry analysis of P-selectin expression (CD62P) and activation of GPIIb/IIIa complex (PAC-1 binding) in whole blood samples, either those stimulated with ADP or collagen as agonists, and in unstimulated controls. Following platelet activation, the GPIIb/IIIa receptor undergoes conformational changes to reveal a ligand binding site specific for fibrinogen, among others. This site is vital for promoting blood platelet aggregation, which is recognized by PAC-1. In the present study, PAC-1 expression was found to be inhibited by exposure to the tested plant fractions, especially fraction A; this fraction has also previously been found to significantly lower blood platelet aggregation in washed blood platelets and platelet rich plasma [10]. Taken together, these findings suggest that inhibition of platelet aggregation may be associated with low expression of GPIIb/IIIa.

During platelet activation by agonists such ADP or collagen, blood platelets are known to release α-granules. The membranes of these α-granules include an adhesive protein called P-selectin, which is translocated to the surface during platelet activation. In the present studies, P-selectin expression was found to decrease in the presence of the tested fractions (A–F); however, this process was not always statistically significant.

Previous studies indicate that changes in blood platelet proteomes are often associated with the presence of cardiovascular disease [13]. However, no such change was observed in any of the samples treated with the tested plant fractions (A–F). In addition, blood platelet activation by ADP is known to be mediated by two receptors: P2Y1 and P2Y12. For example, P2Y12 receptor inhibition enhances VASP phosphorylation, a stage in platelet aggregation, whereas its activation is associated with VASP non-phosphorylation [14]. VASP phosphorylation assay is often used to study the interaction between anti-platelet drugs such as clopidogrel and the P2Y12 receptor [15]. Our present findings indicate that the anti-platelet potential of the tested plant fraction is not dependent on the P2Y12 receptor, i.e., no changes in VASP phosphorylation were observed.

In conclusion, both our present findings and those of previous studies [10,16] indicate that fractions isolated from various sea buckthorn organs, especially fraction A in the present study, demonstrate significant potency against platelet hyperactivation; however, their anti-platelet potential does not appear to act through the P2Y12 receptor. In addition, the high anti-platelet activity demonstrated by fraction A may be due to the presence of flavonol glycosides in the fraction, which may also be responsible for its antioxidant activity [17]. In addition, previous findings indicate that fraction A has anti-coagulant and anti-platelet properties, which were observed in washed blood platelets [4,7]. For example, the phenolic fraction taken from fruits (fraction A) prolonged thrombin time and inhibited platelet adhesion in vitro, and changed the level of thiol groups in platelet proteins. Such action may be associated with the presence of flavonol glycosides [4]. A novel finding of this study is that fraction A, similarly to a commercial extract from the berries of *A. melanocarpa* (Aronox^®^); a known source of anthocyanins with different biological activities [17] has anti-platelet properties.

## 4. Materials and Methods

### 4.1. Chemicals

Flow cytometry reagents were purchased from Becton Dickinson (1329 W US-6, Holdrege, NE 68949, USA), PLT VASP/P2Y12 kit was acquired from BioCytex (140 Chemin de l’Armée d’Afrique, 13010 Marseille, France). The PL microchips and other equipment needed for the T-TAS equipment were purchased from Bionicum (Chełmska 21, 00-724 Warszawa, Poland). All reagents necessary for electrophoresis were provided by commercial suppliers, including BIO-RAD (Przyokopowa 33, 01-208 Warszawa, Poland), POCh (Gen. Sowińskiego 11, 44-121 Gliwice, Poland) and Sigma-Aldrich (2033 Westport Center Dr, St. Louis, MO 63146, USA). ADP was obtained from Chrono-Log Corporation (2 W Park Road, Havertown, PA 19083, USA). Collagen type I and dimethylsulfoxide (DMSO) were purchased from Sigma-Aldrich.

A stock solution of commercial product–Aronox (by Agropharm Ltd., Poland; batch No 020/2007k, *Aronia melanocarpa* berry extract) was prepared in H_2_O at a concentration of 5 mg/mL.

### 4.2. Plant Material

Sea buckthorn (*Elaeagnus rhamnoides* (L.) A. Nelson; syn. *Hippophae rhamnoides* L.) organs were obtained from a horticultural farm in Sokółka, Podlaskie Voivodeship, Poland (August, 2015) (53°24′ N, 23°30′ E), the biggest Polish plantation of sea buckthorn. Individual organs (fruits, leaves, and twigs) were transported to the Institute of Soil Science and Plant Cultivation–State Research Institute in Puławy, Poland.

### 4.3. Chemical Characteristics of Fractions from Sea Buckthorn Fruits, Twigs, and Leaves

Stock solutions of the tested fractions were made in 50% DMSO (*v/v*%). The final concentration of DMSO in samples was lower than 0.05% (*v/v*%) and its effect was determined in all experiments. Both phenolic and nonpolar fractions were obtained from sea buckthorn fruits, leaves, and twigs, and these were used for testing. The chemical composition of the tested fractions was determined using a Thermo Ultimate 3000RS UHPLC system (USA). Regarding the sea buckthorn fruits, the phenolic fractions are dominated by a range of flavonol glycosides, mainly isorhamnetin glycosides and acylated glycosides, while the nonpolar fractions demonstrate high levels of triterpenoids and acylated triterpenoids. For the leaves, the phenolic fraction contains a number of hydrolysable tannins and ellagic acid, together with both non-acylated and acylated flavonol glycosides, as well as triterpenoid saponins; in contrast, the nonpolar fraction is a source of triterpenoids and triterpenoid saponins. Finally, for the twigs, the phenolic fraction contains high levels of proanthocyanidins and catechin, as well as spermidine derivatives, which are acylated with coumarin and ferulic acid; the nonpolar extracts contain high levels of triterpenoids and acylated triterpenoids [4,10]. The content of dominant compounds in the tested fractions, expressed as a percentage of the total peak area, are given in Table 2.

### 4.4. Blood and Blood Platelets Samples

Fresh human blood was collected from healthy volunteers (4 women, 4 men, the average age 28), who were not smoking or taking any drugs, including anti-platelet drugs and supplements at the time. The blood was collected in tubes with CPDA_1_ anticoagulant (citrate/phosphate/dextrose/adenine; 8.5:1; *v/v*; blood/CPDA) (Sigma-Aldrich). The biological material was made available by the L. Rydygiera Medical Center in Łódź. The study was conducted with the consent of the local Bioethical Committee (UŁ3/KBBN-UŁ/II/2016, 12/10/2016).

The whole blood or washed blood platelets were incubated for 30 min at 37 °C with the tested fractions (final concentration: 5 and 50 µg/mL). Platelets were obtained by centrifugation (1200 rpm, 15 min, 25 °C) followed by suspension in Barber buffer (0.14 M NaCl, 0.014 M Tris, 5 mM glucose, pH 7.4).

### 4.5. Flow Cytometry Analysis

Changes in the platelet activation process were observed using an LSR II Flow Cytometer (Becton Dickinson, San Diego, CA, USA). Whole blood (150 µL) was incubated with extracts and fractions for 30 min at 25 °C. The samples were gently vortexed. After 15 min, the samples were treated with platelet agonists: ADP at final concentrations of 10 and 20 µM, or collagen at a final concentration of 10 µg/mL. The samples were then diluted 10-fold in sterile PBS with Mg^2+^, and 3 µL of antibodies (CD61/PerCP; CD62/PE and PAC-1/FITC) were added to the cytometric tubes. The antibodies were then prepared for compensation settings and for isotype controls (CD61/PerCP, FITC isotype, PE isotype), following which, the samples were transferred to cytometric tubes and gently vortexed.

The samples were labelled in the dark for 30 min at 25 °C. The platelets were fixed in CellFix and incubated for one hour at 37 °C. All samples were vortexed before measurement. The obtained results were analysed using FACSDiva software [18,19].

### 4.6. Platelet VASP Phosphorylation

The specific ADP receptor for platelets (P2Y12) was monitored using a flow cytometry kit (PLT VASP/P2Y12). The test was carried out according to the manufacturer’s instructions. The results are presented in the form of PRI (platelet reactivity index) [20].

### 4.7. Polyacrylamide gel Electrophoresis Analysis

In the first stage, cell lysates were prepared. Whole blood was centrifuged (1200 rpm, 15 min, 25 °C). The resulting platelet rich plasma was centrifuged as above. Low platelet plasma was collected and frozen. The platelets were suspended in Barber buffer. The plate suspension was incubated with test extracts and fractions for 30 min at 37 °C. After the incubation, the samples were centrifuged as above, and lysis buffer was added to the pellet. Samples were sonicated and centrifuged (5000 rpm, 5 min, 25 °C). The supernatant was transferred to new Eppendorf tubes. Protein separation was carried out under reducing and non-reducing conditions [21].

### 4.8. Total Thrombus-Formation Analysis System (T-TAS)

Platelet plug formation was measured in real time under blood flow conditions. Briefly, whole blood (400 µL) was incubated with test extracts and fractions, and 320 µL of blood was then transferred to the reservoir. Plug formation was determined using a PL chip based on the AUC_10_ (Area Under the Curve) parameter [22].

### 4.9. Data Analysis

The Q-Dixon test was performed to eliminate uncertain data. All the values in this study were expressed as mean ± SD; n—number of blood donors. Statistical analysis was performed with one-way ANOVA for repeated measurements.

## Figures and Tables

**Figure 1 ijms-22-03282-f001:**
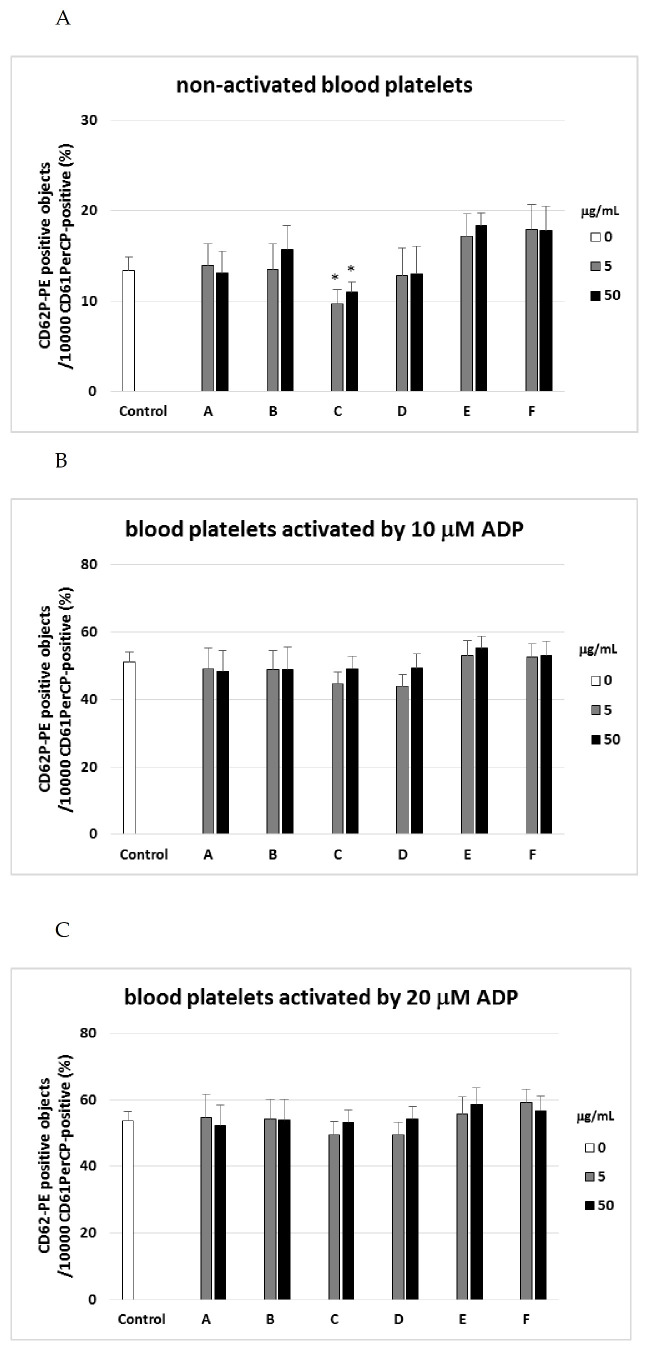
Effects of different plant fractions (5 and 50 µg/mL; 30 min) on expression of P-selectin on resting (**A**) or agonist-stimulated blood platelets: 10 µM ADP (adenosine diphosphate) (**B**), 20 µM ADP (**C**), and 10 µg/mL collagen (**D**) in whole blood samples. The blood platelets were distinguished based on the expression of CD61/PerCP. For each sample, 10,000 CD61-positive objects (blood platelets) were acquired. For the assessment of P-selectin expression, samples were labeled with fluorescently conjugated monoclonal antibody CD62P. Results are shown as the percentage of platelets expressing CD62P. Data represent mean ± SD of 6 healthy volunteers (each experiment performed in triplicate). * *p* < 0.05 (vs. control platelets–blood platelets without tested fraction).

**Figure 2 ijms-22-03282-f002:**
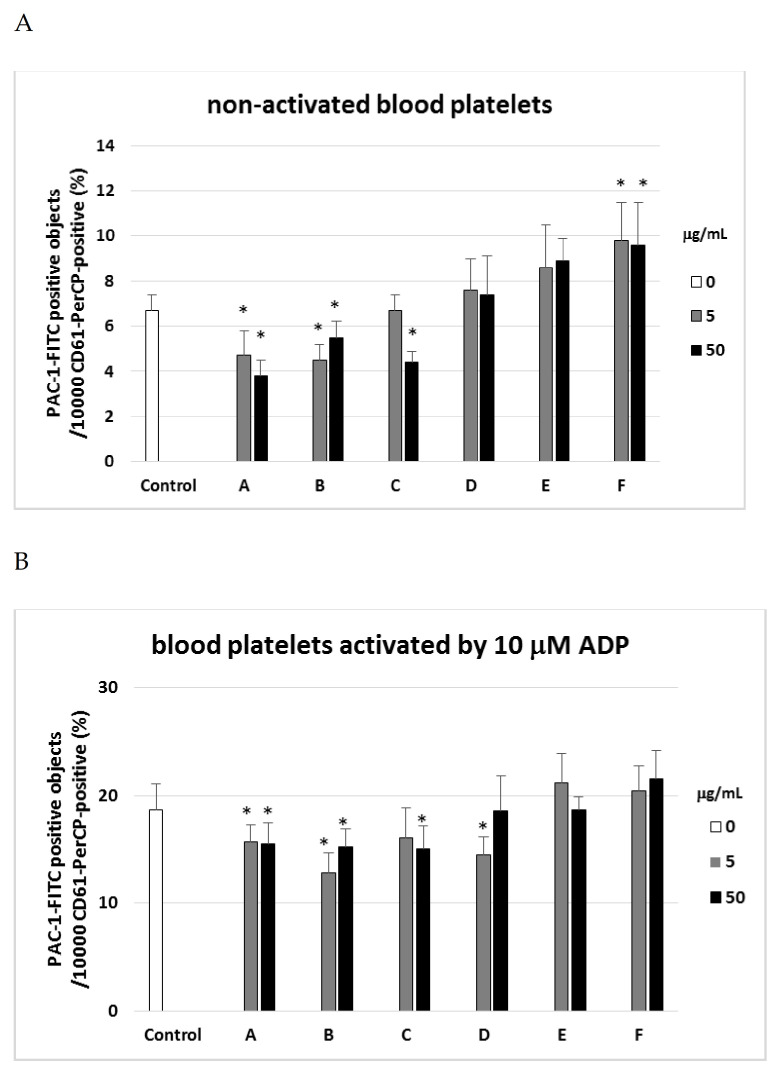
Effects of different plant fractions (5 and 50 µg/mL; 30 min) on expression of the active form of GPIIb/IIIa on resting (**A**) or agonist-stimulated blood platelets: 10 µM ADP (**B**), 20 µM ADP (**C**), and 10 µg/mL collagen (**D**) in whole blood samples. The blood platelets were distinguished based on the expression of CD61. For each sample, 10,000 CD61-positive objects (blood platelets) were acquired. For the assessment of GPIIb/IIIa expression, samples were labeled with fluorescently conjugated monoclonal antibody PAC-1/FITC. Results are shown as the percentage of platelets binding PAC-1/FITC. Data represent mean ± SD of 6 healthy volunteers (each experiment performed in triplicate). * *p* < 0.05 (vs. control platelets–blood platelets without tested fraction).

**Figure 3 ijms-22-03282-f003:**
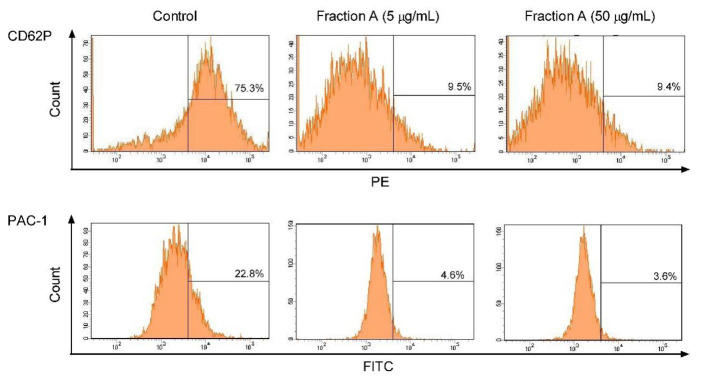
Effects of fraction A (concentration 5 and 50 µg/mL, incubation time—30 min) on the expression of P-selectin and the active form of GPIIb/IIIa on platelets stimulated by 10 µg/mL collagen in whole blood samples. Figure demonstrates selected diagrams.

**Figure 4 ijms-22-03282-f004:**
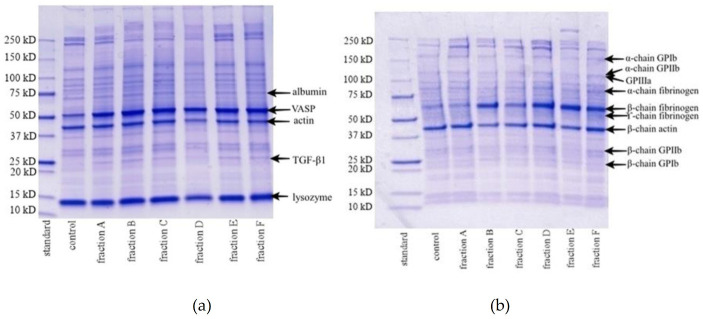
Electrophoretic patterns of blood platelet proteome in the presence of different plant fractions (50 µg/mL; 30 min): reducing conditions (**a**), non-reducing conditions (**b**).

**Figure 5 ijms-22-03282-f005:**
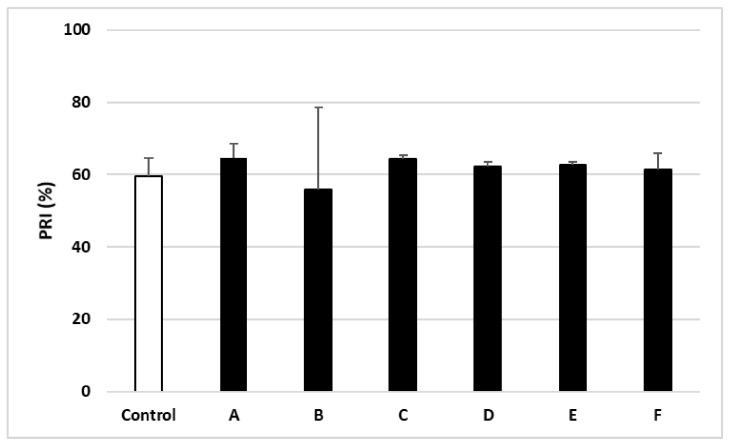
Effects of different plant fractions (50 µg/mL; 30 min) on vasodilator-stimulated phosphoprotein (VASP) phosphorylation in ADP—activated blood platelets. Data represent mean ± SD of 6 healthy volunteers (each experiment performed in triplicate).

**Figure 6 ijms-22-03282-f006:**
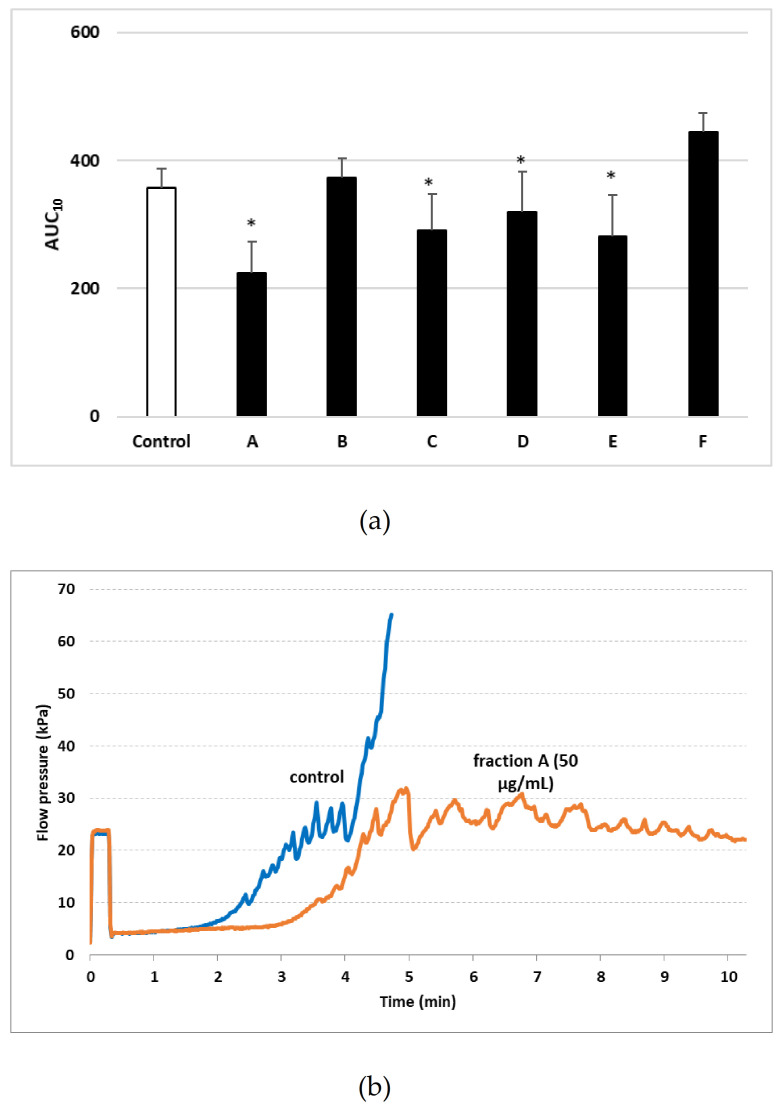
Effects of different plant fractions (50 µg/mL; 30 min) on the T-TAS (Total Thrombus formation Analysis system) using the PL-chip (chip for analysis of platelet thrombus formation (primary hemostatic ability)) in whole blood samples (**a**). Whole blood samples were analyzed by the T-TAS at the shear rates of 1000 s^−1^ on the PL-chips. Area under the curve (AUC_10_) in PL are shown as closed circles. Data represent mean ± SD of 6 healthy volunteers (each experiment performed in triplicate). * *p* < 0.05 (vs. control sample–whole blood without tested fraction). Figure 6 (**b**) demonstrates selected diagram for fraction A.

**Table 1 ijms-22-03282-t001:** A comparison of the effects of the phenolic fractions and the nonpolar fractions isolated from various organs of sea buckthorn (A–F, tested at 50 µg/mL) and aronia berry extract (50 µg/mL) on biomarkers of platelet activation measured by cytometric analysis and T-TAS

Fraction	CD62P Expression				PAC-1 Expression				T-TAS	VASP Phosphorylation
	Non-Activated Platelets	Platelets Activated by 10 µM ADP	Platelets Activated by 20 µM ADP	Platelets Activated by 10 µg/mL Collagen	Non-Activated Platelets	Platelets Activated by 10 µM ADP	Platelets Activated by 20 µM ADP	Platelets Activated by 10 µg/mL Collagen		
A	No effect	No effect	No effect	No effect	Decrease (anti-platelet potential)	Decrease (anti-platelet potential)	No effect	Decrease (anti-platelet potential)	Anti-coagulant poteintial	No effect
B	No effect	No effect	No effect	No effect	Decrease (anti-platelet potential)	Decrease (anti-platelet potential)	No effect	No effect	No effect	No effect
C	Decrease (anti-platelet potential)	No effect	No effect	No effect	No effect	No effect	No effect	Decrease (anti-platelet potential)	Anti-coagulant poteintial	No effect
D	No effect	No effect	No effect	No effect	No effect	No effect	No effect	No effect	Anti-coagulant poteintial	No effect
E	No effect	No effect	No effect	No effect	No effect	No effect	No effect	Decrease (anti-platelet potential)	Anti-coagulant poteintial	No effect
F	No effect	No effect	No effect	No effect	Increase (pro-activation potential)	No effect	Increase (pro-activation potential)	No effect	No effect	No effect
Aronia berry extract	No effect	Decrease (anti-platelet potential)	Decrease (anti-platelet potential)	Decrease (anti-platelet potential)	No effect	No effect	Decrease (anti-platelet potential)	Decrease (anti-platelet potential)	No effect	No effect

**Table 2 ijms-22-03282-t002:** The content of dominant compounds in the tested fraction expressed as a percentage of the total peak area.

**Fractions**	**Relative Peak Area %**
Polyphenols rich fraction of sea buckthorn from fruits (fraction A)	
Flavonol glycosides, non-acylated and acylated	67.1
Triterpenoids and acylated triterpenoids	9.1
Triterpenic acids rich fraction of sea buckthorn fruits (fraction B)	
Triterpenoids and acylated triterpenoids	83.6
Flavonol glycosides, non-acylated and acylated	0.9
Polyphenols rich fraction of sea buckthorn from leaves (fraction C)	
Hydrolysable tannins and ellagic acid	31.3
Flavonol glycosides, non-acylated and acylated	24.5
Triterpenic acids rich fraction of sea buckthorn leaves (fraction D)	
Triterpenoids and acylated triterpenoids	50.7
Triterpenoid saponins	30.5
Polyphenols rich fraction of sea buckthorn from twigs (fraction E)	
Proanthocyanidins and catechin	54.3
Spermidine derivatives	10.7
Triterpenic acids rich fraction of sea buckthorn from twigs (fraction F)	
Triterpenoids and acylated triterpenoids	89.0
Catechin and proanthocyanidins	1.3

## Data Availability

Data sets used and/or analysed in this study are available from the corresponding author on reasonable request.

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
