# Peer review of "Anti-Platelet Properties of Phenolic and Nonpolar Fractions Isolated from Various Organs of Elaeagnus rhamnoides (L.) A. Nelson in Whole Blood"

_ijms, 2021, doi:10.3390/ijms22063282_

Round 1

Reviewer 1 Report

The Authors implemented all my suggestions. However, in Figures 1 and 2, they reported µ/mL. Please, correct the text in µg/mL. In my opinion, after these changes, the paper is suitable for publication.

Author Response

The Authors implemented all my suggestions. However, in Figures 1 and 2, they reported µ/mL. Please, correct the text in µg/mL. In my opinion, after these changes, the paper is suitable for publication.

Response: We have corrected Fig. 1 and 2.

Reviewer 2 Report

  1. Authors could increase the citation rate of their publications if they used the generally accepted Latin name of sea buckthorn Hippophae rhamnoides The name Elaeagnus rhamnoides (L.) A. Nelson does not fully reflect the generic affiliation and taxonomic
    position of the plant studied by the authors.
  2. The term nonpolar fractions can only be used conditionally and should be enclosed in quotation marks, since in fact the fraction designated by this term is a concentrate of polyfunctional triterpenoids of an acidic nature. This fraction cannot be called nonpolar, since its dielectric permittivity is quite high. According to the method of isolation, the fractions obtained by defatting of raw material and water-methanol extract with hexane are really non-polar. It contains, in addition to fats, hydrocarbons, aliphatic and triterpene alcohols in free and acylated form, including sterols, polyprenols and tocopherols, esterified aliphatic acids and other bioactive constituents. The studied fractions can more accurately be called flavonoids rich (polyphenols rich) and acidic triterpenoids-rich (titerpenic acids rich).
  3. Authors should make references to literature data more carefully and bring them in line with the instructions for authors. “In the text, reference numbers should be placed in square brackets [ ], and placed before the punctuation; for example [1], [1–3] or [1,3]. For embedded citations in the text with pagination, use both parentheses and brackets to indicate the reference number and page numbers; for example [5] (p. 10). or [6] (pp. 101–105).” Link 12 is literally the same as link 15.
  4. Authors should limit self-citation to the most important publications. Most scientific journals regulate the level of self-citation of 25%.

Author Response

  1. Authors could increase the citation rate of their publications if they used the generally accepted Latin name of sea buckthorn Hippophae rhamnoides The name Elaeagnus rhamnoides (L.) A. Nelson does not fully reflect the generic affiliation and taxonomic
    position of the plant studied by the authors.

Response: According to The Plant List, the correct name for sea buckthorn is Elaeagnus rhamnoides (L.) A. Nelson. We also used this name in our earlier publications.

  1. The term nonpolar fractions can only be used conditionally and should be enclosed in quotation marks, since in fact the fraction designated by this term is a concentrate of polyfunctional triterpenoids of an acidic nature. This fraction cannot be called nonpolar, since its dielectric permittivity is quite high. According to the method of isolation, the fractions obtained by defatting of raw material and water-methanol extract with hexane are really non-polar. It contains, in addition to fats, hydrocarbons, aliphatic and triterpene alcohols in free and acylated form, including sterols, polyprenols and tocopherols, esterified aliphatic acids and other bioactive constituents. The studied fractions can more accurately be called flavonoids rich (polyphenols rich) and acidic triterpenoids-rich (titerpenic acids rich).

Response: We have changed. Now, we have used “polyphenols rich fraction” and “triterpenic acids rich fraction”.

  1. Authors should make references to literature data more carefully and bring them in line with the instructions for authors. “In the text, reference numbers should be placed in square brackets [ ], and placed before the punctuation; for example [1], [1–3] or [1,3]. For embedded citations in the text with pagination, use both parentheses and brackets to indicate the reference number and page numbers; for example [5] (p. 10). or [6] (pp. 101–105).” Link 12 is literally the same as link 15.

Response: We have corrected.

  1. Authors should limit self-citation to the most important publications. Most scientific journals regulate the level of self-citation of 25%.

Response: We have reduced the number of self-citations. Other self-citations are necessary.

This manuscript is a resubmission of an earlier submission. The following is a list of the peer review reports and author responses from that submission.

Round 1

Reviewer 1 Report

In this study, the authors would like to identify changes in blood platelet proteomes following treatment with sea buckthorn fractions A-F and try to determine anticoagulant and anti-platelet properties of Elaeagnus rhamnoides (L.) A. Nelson.

  1. In this study, the blood samples were collected from healthy volunteers. How many healthy volunteers in each test? The author did not describe the detail of clinical samples. The authors need to provide the basic information (ex: age, gender…) of these volunteers.

  1. In the result section, the authors only make the brief description in figure 3 and I cannot find the detailed information for figure 3.

  1. Fraction A did not affect the CD62P expression in fig 1 and table 2, but fraction A suppresses CD62P in fig 3. The results are not consistent.

  1. The author would like to demonstrate that the function of the anticoagulant of Elaeagnus rhamnoides (L.) A. Nelson. Therefore, in the experimental design, they need to provide a positive control (clinical antiplatelet agents, ex: aspirin, abciximab, tirofiban) to demonstrate that the anti-platelet property can be detected by their models.

  1. The authors should reorganize the whole content of this manuscript because it is difficult to read and hard to understand the author’s description.

Author Response

In this study, the authors would like to identify changes in blood platelet proteomes following treatment with sea buckthorn fractions A-F and try to determine anticoagulant and anti-platelet properties of Elaeagnus rhamnoides (L.) A. Nelson.

  1. In this study, the blood samples were collected from healthy volunteers. How many healthy volunteers in each test? The author did not describe the detail of clinical samples. The authors need to provide the basic information (ex: age, gender…) of these volunteers.

Response:  We have added this information.

  1. In the result section, the authors only make the brief description in figure 3 and I cannot find the detailed information for figure 3.

 Response: All relevant information is included in the description.

  1. Fraction A did not affect the CD62P expression in fig 1 and table 2, but fraction A suppresses CD62P in fig 3. The results are not consistent.

Response:  The results indicated altered blood platelet activation states in all samples treated with tested plant fractions (A-F) compared with untreated controls; this was true for both samples treated with agonist (ADP or collagen) and those which were not (Fig. 1-3). However, these changes were not always statistically significant. Moreover, Figure 3 demonstrates selected diagrams for effects of fraction A (concentration 5 and 50 µg/mL, incubation time – 30 min) on the expression of P-selectin and the active form of GPIIb/IIIa on platelets stimulated by 10 µg/mL collagen in whole blood samples.

  1. The author would like to demonstrate that the function of the anticoagulant of Elaeagnus rhamnoides (L.) A. Nelson. Therefore, in the experimental design, they need to provide a positive control (clinical antiplatelet agents, ex: aspirin, abciximab, tirofiban) to demonstrate that the anti-platelet property can be detected by their models.

 Response:  The aim of our present manuscript was only to determine if tested plant fractions have-anti-platelet potential in whole blood using T-TAS and flow cytometry. Moreover, our earlier results demonstrated that commercial product – Aronox (with aronia berries) has also anti-platelet properties (using the same model and flow cytometry) (for example, Olas et al., 2010, Platelets). However, in future manuscripts, we will be present results for typical anti-platelet drugs, including aspirin.

  1. The authors should reorganize the whole content of this manuscript because it is difficult to read and hard to understand the author’s description.

Response: In the chapter of Introduction, we have described the role of anticoagulants, antiplatelet drugs and dietary components in the prevention and treatment of cardiovascular diseases (in the first part). Next, we have described our earlier results that demonstrate the effect of various fractions from sea buckthorn on coagulation and blood platelet functions (using washed platelets and platelet-rich plasma). In the end, we have described the aim of our present experiments (using whole blood).

Reviewer 2 Report

The Authors submitted a manuscript regarding the anti-platelet properties of phenolic and non-polar fractions isolated from various organs of Elaeagnus rhamnoides (L.) A. Nelson in whole blood. The work is of interest and I have just a few suggestions.

line 1: “non-polar” - remove hyphen;

lines 5-12: affiliations – check the font style;

lines 13 and 40: I suggest to report the common name of the plant in brackets;

lines 74: the Authors should mention the month and year of plant collection;

lines 80 and 81: please, specify whether the % is a w/w%, w/v% or v/v%;

line 97 (Table 1): the Authors should report retention time of the fractions;

line 103: the Authors should indicate the company from which they purchased CPDA;

Figures 1 and 2: please, report µg/ml next to 0, 5, 50 and a caption for A-F (detail of the fractions);

Figure 3: image resolution must be improved;

Figures 5 and 6: please, provide a caption for A-F (detail of the fractions);

Table 2: please, simplify the text and provide a caption with abbreviations;

Besides, the conclusion section should be added in the manuscript.

Author Response

The Authors submitted a manuscript regarding the anti-platelet properties of phenolic and non-polar fractions isolated from various organs of Elaeagnus rhamnoides (L.) A. Nelson in whole blood. The work is of interest and I have just a few suggestions.

line 1: “non-polar” - remove hyphen;

Response: It is done.

lines 5-12: affiliations – check the font style;

Response: We have corrected.

lines 13 and 40: I suggest to report the common name of the plant in brackets;

Response: It is done.

lines 74: the Authors should mention the month and year of plant collection;

Response: We have added.

lines 80 and 81: please, specify whether the % is a w/w%, w/v% or v/v%;

Response: We have added.

line 97 (Table 1): the Authors should report retention time of the fractions;

Response: We cannot provide the retention time. Different separation methods have been used.

line 103: the Authors should indicate the company from which they purchased CPDA;

Response: It is done.

Figures 1 and 2: please, report µg/ml next to 0, 5, 50 and a caption for A-F (detail of the fractions);

Response: Concentration were added. Fractions details are described earlier.

Figure 3: image resolution must be improved;

Response: It is done.

Figures 5 and 6: please, provide a caption for A-F (detail of the fractions);

Response: Concentration were added. Fractions details are described earlier.

Table 2: please, simplify the text and provide a caption with abbreviations;

Response: Abbreviations list is on the last page.

Besides, the conclusion section should be added in the manuscript.

Response: We have added the conclusion.

Round 2

Reviewer 1 Report

The authors only minor modified the manuscript without any improvements.  This manuscript is still need English editing. In the result section, the description of the data is too complicate to understand. In the experimental design, I suggest that the author need to provide a positive control to demonstrate that the fraction A indeed have anti-platelet activity same as Aronox (the authors’ previous report) or other clinical drugs. 

Author Response

Response:

We would like to thank the Reviewer for providing helpful comments.

Our paper has been proofread by a professional translator who is a native speaker of English (Adhoc English).

In the result section, the description of the data has been changed.

In the result discussion (in conclusion), we have added information: “A novel finding of this study is that fraction A, similarly to a commercial extract from the berries of Aronia melanocarpa (Aronox®; a known source of anthocyanins with different biological activities (Olas et al., 2010) has anti-platelet properties.”